# An Improved Clustering Algorithm for Multi-Density Data

Abdulwahab Ali Almazroi * and Walid Atwa

Department of Information Technology, College of Computing and Information Technology at Khulais, University of Jeddah, Jeddah 23451, Saudi Arabia
* Correspondence: aalmazroi@uj.edu.sa

**Abstract:** The clustering method divides a dataset into groups with similar data using similarity metrics. However, discovering clusters in different densities, shapes and distinct sizes is still a challenging task. In this regard, experts and researchers opt to use the DBSCAN algorithm as it uses density-based clustering techniques that define clusters of different sizes and shapes. However, it is misapplied to clusters of different densities due to its global attributes that generate a single density. Furthermore, most existing algorithms are unsupervised methods, where available prior knowledge is useless. To address these problems, this research suggests the use of a clustering algorithm that is semi-supervised. This allows the algorithm to use existing knowledge to generate pairwise constraints for clustering multi-density data. The proposed algorithm consists of two stages: first, it divides the dataset into different sets based on their density level and then applies the semi-supervised DBSCAN algorithm to each partition. Evaluation of the results shows the algorithm performing effectively and efficiently in comparison to unsupervised clustering algorithms.

**Keywords:** clustering; DBSCAN; pairwise constraints; multi-density data

## 1. Introduction

Clustering is utilized to arrange a dataset into a limited set of clusters based on selected similarity metrics [1]. There are different clustering categories, such as density-based algorithms that can identify clusters of distinct sizes and shapes. Hence, this algorithm does not require the specification of cluster numbers—rather, clusters are identified in a densely connected region that grows based on the direction of density, meaning that density-based algorithms identify clusters based on regions that have a high density and are separated from regions with low densities [2–5].

The DBSCAN algorithm provides all the advantages of the density-based clustering family [6]. It computes the density by counting the number of points within a circle with a given radius (referred to as *Eps*) surrounding the point. Core points are characterized as having a density over a predetermined threshold (called *MinPts*) [6]. The two provided parameters (*Eps* and *MinPts*) define a single density. Thus, clustering methods based on the DBSCAN cannot perform well with multi-density data [7,8]. Moreover, most of these methods are unsupervised and cannot utilize prior knowledge, which may be available either in the form of labeled data or pairwise constraints.

Semi-supervised clustering improves clustering performance with pairwise constraints or labeled data. Pairwise constraints are of two types: *must-link* and *cannot-link* constraints. A *must-link* constraint (i.e., $ML(x, y)$) means that the two objects $x$ and $y$ must be in the same cluster. Meanwhile, the *cannot-link* constraint (i.e., $CL(x, y)$) means that these two objects $x$ and $y$ must be in different clusters.

In this study, we present a semi-supervised clustering algorithm for clustering multi-density data, referred to as the SSMD. Our algorithm is separated into two core parts: Firstly, we divided the data into various density levels and calculate the density parameters

for each density level set. Secondly, we applied pairwise constraints to get a set of clusters based on the computed parameters. We conducted experiments with real datasets. The comparison results reveal the effectiveness and efficiency of the proposed algorithm.

The main contributions are twofold: Firstly, our proposed algorithm can discover clusters of changing densities using the available pairwise constraints. Secondly, we evaluated the performance of the proposed algorithm on different experiments with varied datasets in comparison to other algorithms.

The remaining part of the paper is organized as follows: Related work is explained in Section 2. In Section 3, the SSMD algorithm is discussed in detail. Section 4 shows the performance analysis of the SSMD algorithm compared with other algorithms. Lastly, the paper is concluded in Section 5.

## 2. Related Work

In this section, the key literature related to clustering methods is highlighted and discussed. Hence, the literature on density-based clustering and semi-supervised clustering algorithms is critically reviewed.

For the purpose of finding clusters in huge spatial data sets, Ester et al. proposed a density-based clustering algorithm named DBSCAN [6]. The DBSCAN algorithm depends on two specified parameters (*Eps* and *Minpts*) that define a single density. Thus, the DBSCAN cannot cluster datasets with large differences in densities well.

Ankerst et al. proposed an algorithm named OPTICS that executes and stores dual parameters—the core distance and reachability distance—for cluster identification with different densities [9]. If the *Eps*-neighborhood of a point *p* contains at least *MinPts* points, then that point is a core point. As a result, a core distance is assigned to each point, describing how far it is from the *MinPts-th* closest point. The greater of the distances between two points—*o* and *p*—or *p*'s core distance, defines the reachability distance between them. The algorithm further creates an ordering process for the dataset which will represent its density-based clustering structure. This study revealed a process which did not extract both traditional clustering information and intrinsic clustering structure automatically with good efficiency [9].

Based on the issue of identifying clusters in high-dimensional data, Ertoz et al. proposed an algorithm that identifies clusters with distinct sizes, shapes, and densities [10]. Firstly, the algorithm checks for the nearest neighbors (*NN*) of individual data points and further describes the similarity that resides between the points with respect to the number of *NN* shared by the points. Hence, the algorithm defines and builds clusters around these defined points. An experiment on various datasets showed that the algorithm achieved good performance compared to traditional methods. To detect clusters of distinct shapes and sizes, Liu et al. proposed an altered version of the DBSCAN algorithm, coining the term Entropy and Probability Distribution (EPDCA) [11] for it. Testing on benchmark datasets indicated that clustering results based on EPDCA achieve good performance.

A study by Kim et al. proposed a density-based clustering algorithm, coined approximate adaptive AA-DBSCAN [12]. The algorithm focused on minimizing extra computation needed to determine parameters by utilizing e-distance for each density when identifying clusters. Based on an experiment conducted, the result showed a significant improvement with regards to clustering performance, with a decrease in running time. Another study by Zhang et al., proposed GCMDDBSCAN because DBSCAN cannot handle databases that are large [13]; the authors highlighted that clustering capabilities on datasets that are large were improved, together with clustering accuracy.

FlockStream was proposed by Forestiero et al. This algorithm is based on a multi-agent system and the results demonstrate that FlockStream shows good performance on both synthetic and real datasets [14]. Chen and Tu proposed a framework named D-Stream with the aim of clustering stream data by utilizing a density-based approach [15].

Recently, semi-supervised clustering algorithms [16–22] have been created as an extension of known unsupervised clustering algorithms that utilize background knowledge in

the form of labeled data or pairwise constraints to improve clustering performance. Huang et al. proposed an algorithm named MDBSCAN that handles multi-density datasets by automatically calculating the parameter *Eps* for each density distribution using pairwise constraints [16].

A study by Ruiz et al. proposed a pairwise-constrained clustering algorithm (C-DBSCAN) that utilizes information pairwise constraints to enhance clustering performance [17]. C-DBSCAN creates a set of neighborhoods based on cannot-link constraints and uses must-link constraints to merge the local clusters in each neighborhood. The results revealed that C-DBSCAN can detect arbitrary shapes and evolving clusters with current pairwise constraints. In another study by Lelis et al., a new density-based semi-supervised clustering algorithm was proposed (SSDBSCAN) that utilizes labeled data for the evaluation of density parameters [18]. The results showed an improvement in clustering accuracy with little supervision required. However, C-DBSCAN and SSDBSCAN cannot be applied well to multi-density data—especially with increases in the number of pairwise constraints.

Wagstaff et al. proposed a well-known semi-supervised clustering algorithm named MPCKmean. The MPCKmean is a variant of the *K*-means algorithm, which uses pairwise constraints for clustering [19]. It is very effective at processing huge datasets, but it cannot handle clusters of different sizes and densities.

### 3. The SSMD Algorithm

Semi-supervised clustering algorithms utilize a set of class label constraints on some examples to help unsupervised clustering. We propose an active semi-supervised clustering method for a multi-density dataset (called the **SSMD**) that attempts to identify clusters with distinct densities and arbitrary shapes. Let *D* be a dataset of *n* points in a *d*-dimensional space. Each point $pi$, is given by $p_i = \{p_{i1}, p_{i2}, \dots, p_{id}\}$. Some supervision information is available in the form of must-link constraints $ML = (p_i, p_j)$ or cannot-link constraints $CL = \{(p_i, p_j)$ where $p_i$ and $p_j$ belong to the same class or different classes, respectively. The *SSMD* algorithm has two main components: (1) Partition the dataset *D* into distinct density level sets. (2) Apply the *Semi-DBSCAN* algorithm to each density level set, where *Semi-DBSCAN* is an extension of *DBSCAN* that makes full use of existing pairwise constraints. The pseudocode of our *SSMD* is revealed in Algorithm 1.

First, we computed the density function for each data point (*p*), as shown in Equation (1). Then, the density for all the data points was sorted in descending order to evaluate the variation between each sequence point. The density variation explained how much denser or sparser the points were:

$$Den(p, MinPts) = \frac{kdist(p, k)}{k} . \tag{1}$$

We defined *kdist*(*p*, *k*) as the number of *p*'s neighbors and *k* as an arbitrary positive integer that explains the *k*-nearest neighbor distance. The density variation of point $p_i$ with respect to $p_j$ is computed in Equation (2):

$$DenVar(p_i, p_j) = \frac{\left| Den(p_i, k) - Den(p_j, k) \right|}{Den(p_i, k)} . \tag{2}$$

We partitioned the dataset into a list of density level sets (*DLS*). The *DLS* contained a set of data points whose densities were approximately the same. We computed the density variation threshold ($\tau$) according to the statistical characteristics of the sorted density variation list:

$$\tau = \frac{\sum_{i=0}^{n} DenVar_{(i)}}{n} \ where \ DenVar_{(i)} > 0. \tag{3}$$

Finally, we separated the data points into different densities based on the computed density variation threshold ($\tau$), as follows:

$$p_i, \ p_j \in DLS_k \ \ if \ DenVar(p_i, \ p_j) \leq \tau. \tag{4}$$

After the density level partitioning, we acquired a list of density level sets (in short, the *DLSList*). Then, we needed to find representative *Eps* for each density level set. We computed the *Epsi* for the *DLS$_i$* as follows:

$$Eps_i = max(DLS_i) \cdot \sqrt{\frac{median(DLS_i)}{mean(DLS_i)}} \ . \tag{5}$$

We defined *max*, *median*, and *mean* as the maximum, median, and mean values of each *DLS$_i$*, respectively.

---

**Algorithm 1 SSMD**

---

**Input**:
   Dataset (*D*), must-link and cannot-link constraints (*ML*, *CL*), number of objects in a neighborhood (*MinPts*)
**Output**:
   Set of clusters
**Begin**
   Compute density value for all data points according to Equation (1)
Sort density list in descending order
   Compute Density variation values using Equation (2)
   Generate Density Level Set (*DLS*) using a threshold in Equation (3)
   *EpsList = EstimateEps(DLSList);*
   **For** each *DLS$_i$* in *DLSList*
   Semi-DBSCAN(*DLS$_i$*, *ML*, *CL*, *MinPts*, *Eps$_i$*)
**End For**
   Return all clusters
**End**

---

With the density level partitioning and parameter estimation completed, we carried on the clustering process: The global parameter *MinPts = k* was initialized and the *Semi-DBSCAN* algorithm adopted for each *Eps$_i$* in the *EpsList* in the corresponding *DLS*. The *semi-DBSCAN* algorithm is able find a set of clusters and a (probably empty) set of noise. Initially, constraints are preprocessed; constraints given by human experts may be incomplete. Some constraints are not explicitly given; for example, if we have a *ML($p_i, p_j$)* and *ML($p_i, p_k$)*, then we have *ML($p_j, p_k$)*. Then, we used constraints to monitor the process of growing clusters in Semi-DBSCAN, as presented in Algorithm 2.

First, we computed the neighborhood for each unclassified data point and compared it with *MinPts* to determine whether it could be added to the current cluster or noise set. Then, clusters were expanded using pairwise constraints as follows:

1.   Add all data points that have a must-link constraint with *d* to the current cluster.
2.   Add all data points in *d's* neighborhood that do not violate the cannot-link constraint with *d* to the current cluster.

---

**Algorithm 2 Semi-DBSCAN** (*D, ML, CL, MinPts, Eps*)

---

**Begin**
*Cluster.Id*: = 0;
**For** each *d* in *D*
    **If** *d* is *UNCLASSIFIED* then
     Compute *d's Eps*-neighborhood *neighborhood*;
      **If** *neighborhood < MinPts*
       *Add d to NOISE set*;
      **Else**
       *Add d to current cluster* (*ClusterId*);
      **For** each data point *o* that has a must-link constraints *ML*(*d, o*)
       *Add o to current cluster* (*ClusterId*);
      **For** each data point *p* in the neighborhood set that does not violate cannot-link constraints
       *Add p to current cluster* (*ClusterId*);
     *ClusterId = ClusterId + 1*;
**End**

---

## 4. Experimental Results

In this section, the datasets used are presented, followed by the evaluation metrics utilized and the results of the study.

### 4.1. Datasets and Evaluation Metrics

In this section, the utilized datasets are presented. Table 1 outlines the datasets used. For our experiments, real datasets such as Magic and Glass were selected from the UCI repository. Hence, these datasets were labeled with the number of instances, attributes, and cluster labels, as described in Table 1.

**Table 1.** The Data Sets Used in the Experiments.

| Dataset | # Instances | # Attributes | # Clusters |
|---------|-------------|--------------|------------|
| Glass | 214 | 10 | 6 |
| Ecoli | 336 | 8 | 8 |
| Segment | 2310 | 19 | 7 |
| Magic | 19,020 | 10 | 2 |

We utilized four clustering algorithms, i.e., C-DBSCAN [17], SSDBSCAN [18], AA-DBSCAN [12], and MPCKmean [19], along with SSMD to compare their clustering performances. Clustering performance was measured in terms of normalized mutual information (*NMI*) and Clustering Accuracy (*ACC*). *NMI* and Clustering accuracy were measured using Equations (6) and (7):

$$NMI = \frac{I(X;Y)}{(H(X) + H(Y))/2} \, . \tag{6}$$

$$ACC = \frac{\sum_{i=1}^{n} I(\hat{t}_i = t_i)}{n} \, . \tag{7}$$

### 4.2. Performance Analysis

In this sub-section, the performance evaluation based on NMI is presented. This evaluation result was for the proposed algorithm in comparison to the four algorithms SSDBSCAN, AA-DBSCAN, MPCKmeans, and C-DBSCAN. The goal was to find out whether or not the proposed algorithm performed better with varied constraints on distinct datasets. Hence, this experiment presented evidence that the proposed algorithm was more effective in comparison to the compared algorithms.

Figure 1 presents the performance results of the four different datasets (Glass, Ecoli, Segment, and Magic), as presented in Table 1. For each dataset, several constraints were utilized for the NMI.

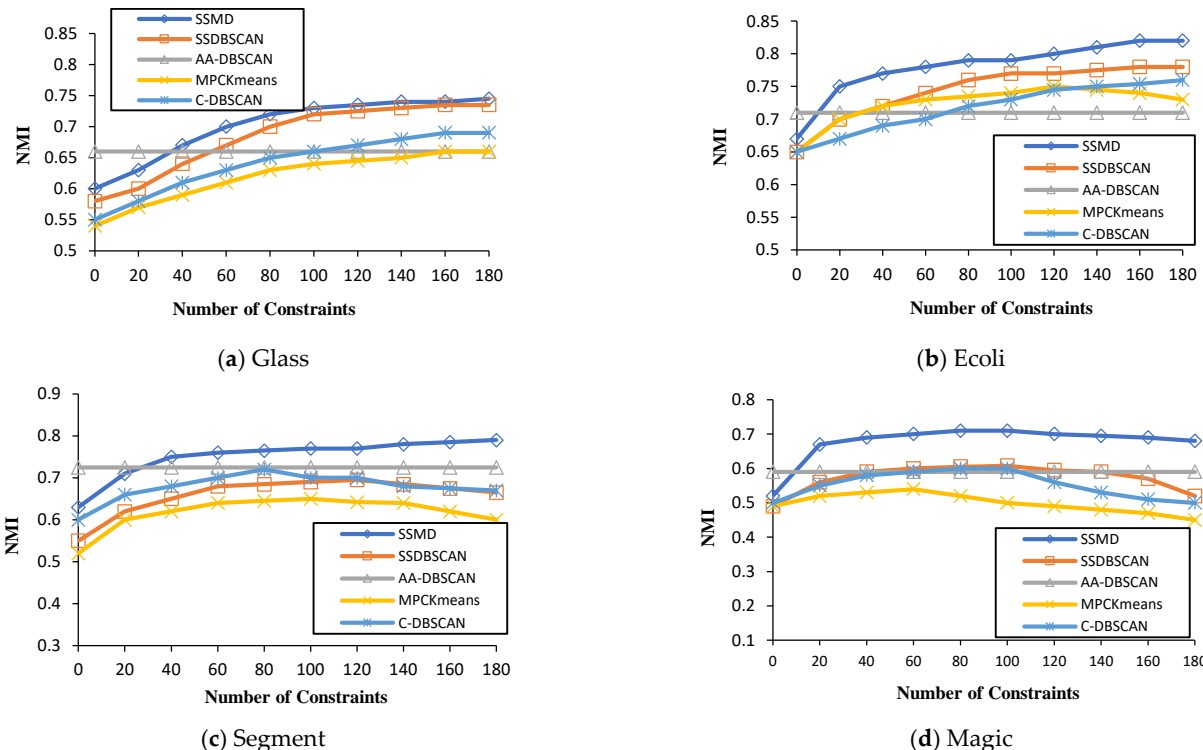

(**a**) Glass

(**b**) Ecoli

(**c**) Segment

(**d**) Magic

**Figure 1.** Comparison of NMI.

It can be observed from Figure 1 that the SSMD algorithm generally performed better than the four other methods when the number of constraints was increased (e.g., ecoli, segment, and magic). For instance, in Figure 1a—with 20 constraints—the proposed algorithm performance was superior, with a greater than 0.6 NMI—and with 80 constraints, the proposed algorithm achieved an NMI of more than 0.7, respectively. Looking very carefully, one can see that SSDBSCAN is the second most effective, followed by C-DBSCAN. It can be seen from Figure 1, that the performance of AA-DBSCAN in all the datasets was at a constant value, as it is an unsupervised clustering algorithm. Thus, this proves the utility of semi-supervised clustering algorithms over unsupervised approaches when knowledge is available.

For the Ecoli dataset, the SSMD algorithm has also shown to be more effective, with a 0.8 NMI with 80 constraints. This was the same with the other two datasets, Segment and Magic. Considering the overall result in Figure 1, we can confidently conclude that for NMI performance evaluation, the proposed algorithm outperformed all four compared algorithms.

Additionally, we present results based on the clustering accuracy of the SSMD algorithm in comparison to other algorithms on the datasets outlined in Table 1. Hence, the results of each algorithm are displayed in varied colors in Figure 2. As seen in Figure 2, the proposed algorithm SSMD had a much higher accuracy and more stable state than SSDBSCAN and C-DBSCAN. For instance, looking at Figure 2a, the accuracy of the SSMD algorithm reached 88% when selecting 20 constraints from all constraints, and reached the highest accuracy when selecting more than 100 constraints.

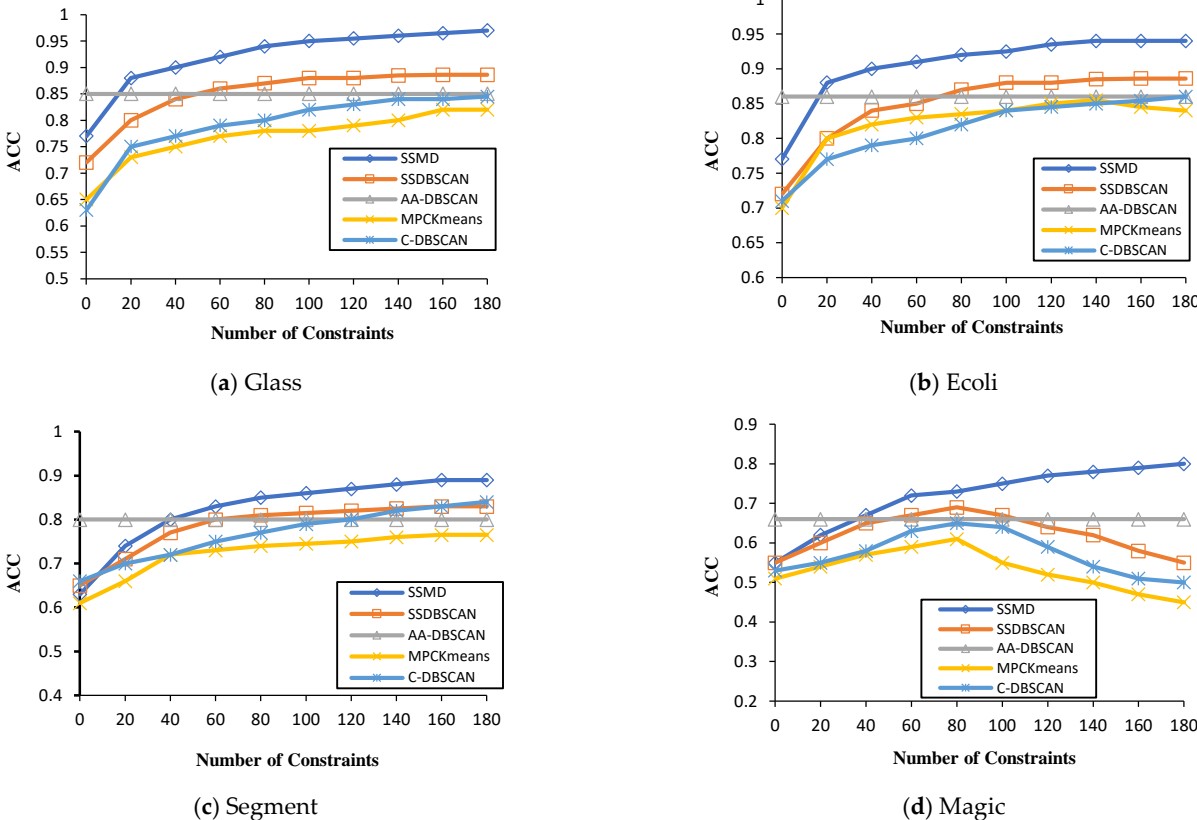

**Figure 2.** Comparison of clustering accuracy.

Considering the general clustering performance on all datasets in Figure 2, SSDBSCAN is the second most effective algorithm for clustering accuracy on the Glass, Ecoli, and Segment datasets, followed by AA-DBSCAN and C-DBSCAN. The main advantage of the proposed algorithm is that it can attain good performance with fewer constraints and progressively maintain performance. However, we observed that the other compared algorithms required more constraints to attain a good performance. Some algorithms' performance also subsided considerably.

### 4.3. Efficiency Analysis Based on Times

In this section, the clustering times for the proposed algorithm and the four compared algorithms are presented. Figure 3 gives the results for the execution times using a distinct number of pairwise constraints on the four datasets. Achieving a low execution time indicates that an algorithm has better performance, and vice versa with a high execution time. From Figure 3a, SSDBSCAN had the highest execution time with all pairwise constraints taken into consideration.

It can be seen that the SSMD method is generally time-efficient (about 5 s on the glass dataset with 214 samples, and about 50 s on the magic dataset with 19,020 samples). Even though the SSMD is not always the fastest method for all of the data sets, it performs substantially better than its competitors.

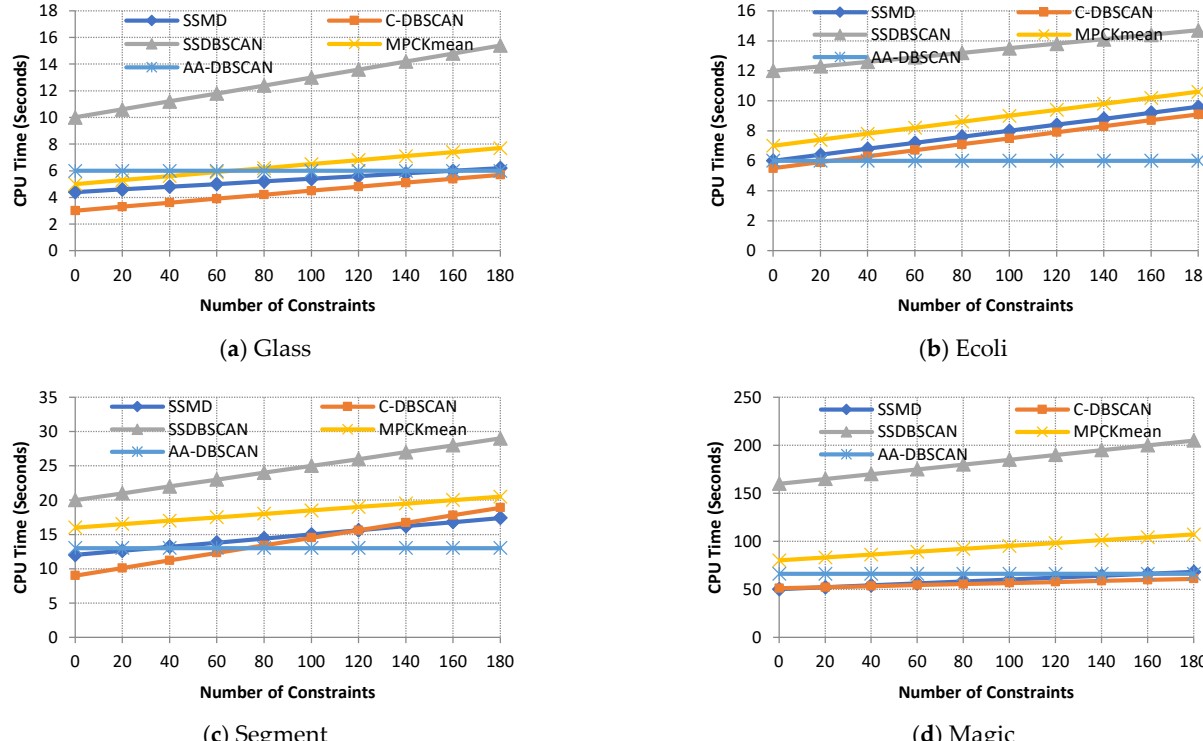

**Figure 3.** Comparison of the execution time.

## 5. Conclusions

In this paper, we proposed the SSMD algorithm, which is a semi-supervised clustering algorithm for clustering multi-density data and arbitrary shapes. The proposed algorithm partitions the dataset into distinct density level sets by examining the statistical characteristics of its variation with respect to density, and then expands the clusters using selected active pairwise constraints. We conducted experiments to assess the clustering performance and execution time on real datasets of distinct dimensions and sizes. The experimental results revealed that the SSMD attained better clustering performance in comparison to the existing state of the art. Furthermore, the SSMD had a significantly reduced the execution time in comparison to the compared algorithms in most scenarios. In the future, we intend to apply the SSMD to the manufacturing environment in order to solve real-world issues and boost production productivity in pertinent industries.

**Author Contributions:** Conceptualization, W.A. and A.A.A.; methodology, W.A.; software, W.A.; validation, A.A.A.; formal analysis, A.A.A.; investigation, W.A.; resources, A.A.A.; data curation, W.A; writing—original draft preparation, W.A.; writing—review and editing, A.A.A.; visualization, W.A.; supervision, A.A.A.; project administration, A.A.A.; funding acquisition, A.A.A. All authors have read and agreed to the published version of the manuscript.

**Funding:** This research was funded by the University of Jeddah, Saudi Arabia, under grant No. (UJ-02-069-DR).

**Institutional Review Board Statement:** Not applicable.

**Informed Consent Statement:** Not applicable.

**Data Availability Statement:** https://archive.ics.uci.edu/ml/datasets.php, accessed on 27 June 2022.

**Acknowledgments:** This work was funded by the University of Jeddah, Saudi Arabia, under grant No. (UJ-02-069-DR). The authors, therefore, acknowledge with thanks the university's technical and financial support.

**Conflicts of Interest:** The authors declare no conflict of interest.

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
