# Peer review of "An Improved Clustering Algorithm for Multi-Density Data"

_axioms, doi:10.3390/axioms11080411_

Round 1

Reviewer 1 Report

1. In the Abstract, it is not clear what "(Eps. and MinPts)" mean.

2. "Furthermore, most existing algorithms are unsupervised methods where available prior knowledge are useless". The clustering is a defined as an unsupervised learning method. Thus, all true clustering algorithm (not most of them) are unsupervised learning methods. Un this paper, the authors propose a semi-supervised algorithm, and this fact must be clearly stated in the Abstract.

3. ". In this study, we focus on using the available knowledge in the form of pairwise constraints to increase the clustering performance. ". It is not clear, which kind of knowledge is used. Are the authors sure that they increased the performance? Probably, the accuracy has been increased? The performance of an algorithm is its speed.

4. A typo: "Hence, this algorithm do not required ..." Must be "does not require"

5. The Introduction is poorly organized. The authors give almost no backgroung and try to explain the peculiarities of their methods without explaining any details of the existing methods. The Intro is too short.

6. In Section 2, the authors also try to explain specific features of various methods without any explanation of the general idea. Many terms such as E-distances or core distance are given without explanation. Probably, it would be better to give some mathematical statements before? In Section 2.2, even for a reader who is familiar with various clustering problems, it is not clear what kind of constraints do the authors mean, and how they are related to "learning similarity metrics". The overview is not systematic.

6. In Section 3, is it a special spelling "Semi-super vised "?

7. "The pseudocode of our SSMD is revealed in algorithm 1." Must be "... in Algorithm 1."

8. In (1), the authors do not explain what MinPts mean, and what p means (while p_i was explained).

9. The punctuation symbols are omitted after all equations.

10. In Algorithm 1, it is not clear what Data set (D), must-link cannot-link constraints (ML, CL) mean, and in which form they myst be given.

11. It seems that Section 2 is not related to Section 3. At least, it was not explained, what is new in Algorithm 1 in comnparison with the existing algorithms. The notation of all used variabls must be checked.

12. "Performance Analysis Based on NMI ". It would be better to avoid using the abbreviations such as NMI in the section headers.

13. The most important flaw of this paper is that the authors do not present any comparison with the existing methods. Why the new method is better than the existing ones? The datased used are comnparatively small.

The novelty of the results must be estimated after improving the way of presenting them.

Author Response

1. In the Abstract, it is not clear what "(Eps. and MinPts)" mean.

Explained in second paragraph in the introduction.

2. "Furthermore, most existing algorithms are unsupervised methods where available prior knowledge are useless". The clustering is a defined as an unsupervised learning method. Thus, all true clustering algorithm (not most of them) are unsupervised learning methods. Un this paper, the authors propose a semi-supervised algorithm, and this fact must be clearly stated in the Abstract.

Explained in the Abstract, and more details in the introduction.

3. ". In this study, we focus on using the available knowledge in the form of pairwise constraints to increase the clustering performance. ". It is not clear, which kind of knowledge is used. Are the authors sure that they increased the performance? Probably, the accuracy has been increased? The performance of an algorithm is its speed.

We use the pairwise constraints as a prior knowledge that improving the clustering performance; (Explained in the third paragraph in Section 1).

The results explained that the proposed algorithm can attain good performance with fewer constraints and progressively maintain the performance. Also, the proposed algorithm get better clustering accuracy than unsupervised (AA-DBSCAN ) and semi-supervised clustering algorithms (SSDBSCAN, C-DBSCAN)
Also, the proposed algorithm
achieving lower execution times on all the datasets in comparison with all the compared algorithms especially compared with SSDBSCAN.

4. A typo: "Hence, this algorithm do not required ..." Must be "does not require"

Yes, Modified.

5. The Introduction is poorly organized. The authors give almost no backgroung and try to explain the peculiarities of their methods without explaining any details of the existing methods. The Intro is too short.

We re-write the introduction and explain more details of the existing methods.

6. In Section 2, the authors also try to explain specific features of various methods without any explanation of the general idea. Many terms such as E-distances or core distance are given without explanation. Probably, it would be better to give some mathematical statements before? In Section 2.2, even for a reader who is familiar with various clustering problems, it is not clear what kind of constraints do the authors mean, and how they are related to "learning similarity metrics". The overview is not systematic.

We re-write Section 2 and explain required comments.

6. In Section 3, is it a special spelling "Semi-super vised "?

Mistake (Modified).

7. "The pseudocode of our SSMD is revealed in algorithm 1." Must be "... in Algorithm 1."

Yes, Modified.

8. In (1), the authors do not explain what MinPts mean, and what p means (while p_i was explained).

We explain MinPts in the second paragraph in the introduction, and explain p in the first paragraph in Section 3 (SSMD Algorithm)

9. The punctuation symbols are omitted after all equations.

Yes, Modified.

10. In Algorithm 1, it is not clear what Data set (D), must-link cannot-link constraints (ML, CL) mean, and in which form they myst be given.

We explain D in the first paragraph in Section 3 (SSMD Algorithm), and must-link cannot-link constraints (ML, CL) explained in the fourth paragraph in the introduction.

11. It seems that Section 2 is not related to Section 3. At least, it was not explained, what is new in Algorithm 1 in comnparison with the existing algorithms. The notation of all used variabls must be checked.

We re-write Section 2 and explain more related algorithms in details.

Our proposed algorithm utilizing the available prior knowledge in the form of pairwise constraint for clustering multi-density data. Also, Experimental results on real datasets demonstrate the effectiveness and efficiency of the proposed algorithm compared with other unsupervised (AA-DBSCAN ) and semi-supervised clustering algorithms (SSDBSCAN, C-DBSCAN)

The notation of all used variables have been checked.

12. "Performance Analysis Based on NMI ". It would be better to avoid using the abbreviations such as NMI in the section headers.

Yes, it's good point (I modified the section headers)

13. The most important flaw of this paper is that the authors do not present any comparison with the existing methods. Why the new method is better than the existing ones? The datased used are comnparatively small.

We compare the proposed algorithm SSMD with three semi-supervised clustering algorithms (SSDBSCAN, C-DBSCAN, and MPCKmean) and one unsupervised clustering algorithm (AA-DBSCAN). The comparisons are evaluated using the Normalized Mutual Information (NMI) and Clustering Accuracy (ACC) Metrics, as shown in Section 4.2.

The results explained that the proposed algorithm can attain good performance with fewer constraints and progressively maintain the performance. Also, the proposed algorithm get better clustering accuracy than unsupervised (AA-DBSCAN ) and semi-supervised clustering algorithms (SSDBSCAN, C-DBSCAN).

Also, we explain the efficiency analysis between proposed algorithm and the other algorithms as shown in Section 4.3.

The proposed algorithm achieving lower execution times on all the datasets in comparison with all the compared algorithms especially compared with SSDBSCAN.

Reviewer 2 Report

Dear Authors,

Thank you for your submission. I think your work has potential but it needs a proper revision. There are some important points to be made:

1. You need re-write your abstract entirely and change the narrative of your paper regarding why and how novel your contribution is. The work needs to be clarified better. From what I see, the proposed method is an improved version of the semi-DBSCAN but named differently which is a bit misleading. In the abstract it gives the impression that the novelty is about using pairwise constraints but almost all your comparisons are with versions of DBSCAN that also use the pairwise constraints. What is then different about your method than the work in the literature? If you clarify this, especially on your abstract, section 2.2 and chapter 3, it would be much more understandable for the readers.

2. The order of references are quite odd and placed confusingly on the narrative of your literature review. There is one instance where the citation is missing (noted on the attached file).

3. The paper needs an extensive editing of English language, supported with proofreading. Please find my comments on the manuscript, attached to this report, pointing out some of these problems. The most common and overly repeated issue is the tense inconsistency. Your alternating between past and present tenses often break the flow of the paper. There are some grammar problems as well like the use of clauses, articles and, etc. 

If you get revisions as a result, I suggest to take your time fully and do not rush.

Thanks

Author Response

Thank you for your submission. I think your work has potential but it needs a proper revision. There are some important points to be made:

1. You need re-write your abstract entirely and change the narrative of your paper regarding why and how novel your contribution is. The work needs to be clarified better. From what I see, the proposed method is an improved version of the semi-DBSCAN but named differently which is a bit misleading. In the abstract it gives the impression that the novelty is about using pairwise constraints but almost all your comparisons are with versions of DBSCAN that also use the pairwise constraints. What is then different about your method than the work in the literature? If you clarify this, especially on your abstract, section 2.2 and chapter 3, it would be much more understandable for the readers.

We re-write the abstract and explain the importance of the proposed method.
also, we improve the introduction and related work sections to explain the
different between our method than the work in the literature.

2. The order of references are quite odd and placed confusingly on the narrative of your literature review. There is one instance where the citation is missing (noted on the attached file).

We ordering the references according to their appearance in the paper

3. The paper needs an extensive editing of English language, supported with proofreading. Please find my comments on the manuscript, attached to this report, pointing out some of these problems. The most common and overly repeated issue is the tense inconsistency. Your alternating between past and present tenses often break the flow of the paper. There are some grammar problems as well like the use of clauses, articles and, etc. 

Thanks very much about your effort and your comments on the manuscript, we apply all the comments and review the paper.

Round 2

Reviewer 1 Report

1. It would be better if the authors describe the data structures in the algorithm with more detaild. E.g. it is not evident that d is a line of data D in Algorithms 1, 2. Array D must be described. The same about ML, CL. Which form of the constraints is used?

2. "For each data point o has a must-link constraints ML(d, o) ". A typo. Must be "For each data point o that has a must-link constraints ML(d, o) "

3. The Conclusion is too short. Authors give almost no discussion where the new algorithm can be used, and which kind of further development of the algorithm they expect.

Author Response

1. It would be better if the authors describe the data structures in the algorithm with more detaild. E.g. it is not evident that d is a line of data D in Algorithms 1, 2. Array D must be described. The same about ML, CL. Which form of the constraints is used?

We explain these in the first paragraph in Section 3

2. "For each data point o has a must-link constraints ML(d, o) ". A typo. Must be "For each data point o that has a must-link constraints ML(d, o) "

Thank you for your comments, we have modified the required comment.

3. The Conclusion is too short. Authors give almost no discussion where the new algorithm can be used, and which kind of further development of the algorithm they expect.

We have modified the conclusion and explain the further development of our study.

Reviewer 2 Report

Dear Authors,

Thank you for carefully revising the paper. I have attached the manuscript with some extra comments based on your revision. 

Author Response

Thank you for carefully revising the paper. I have attached the manuscript with some extra comments based on your revision. 

Thank you for your powerful comments, we have modified the required comments.
